# Investigating the Use Profile of *Kigelia africana* (Lam.) Benth. through Market Survey in Benin

Hubert Olivier Dossou-Yovo [1,*] , Fifanou G. Vodouhè [2] , Valentin Kindomihou [1] and Brice Sinsin [1]

1   Laboratory of Applied Ecology, Faculty of Agronomic Sciences, University of Abomey-Calavi, Abomey-Calavi P.O. Box 526, Benin; vkindomihou@yahoo.fr (V.K.); bsinsin@gmail.com (B.S.)
2   Laboratory of Economic and Social Dynamics Analysis, Faculty of Agronomy, University of Parakou, Parakou P.O. Box 123, Benin; vodouhefifanou@gmail.com
*   Correspondence: dohuoly@yahoo.fr; Tel.: +229-9795-7040

**Abstract:** This research focuses on *Kigelia africana* in Benin where it is widely used in traditional medicine but receives little attention from researchers. In addition, this species has recently been recorded as threatened in the country. The aim was to gather ethnobotanical knowledge using a printed semi-structured questionnaire to collect data from herbal medicine traders, randomly selected, through a face-to-face discussion. The survey was carried out from January to March 2020. Among 36 questioned herbal traders, 36% of respondents obtained parts of *K. africana* by purchase in their own markets and by travelling far (3–10 km covered). The same proportion travel very far before buying parts (more than 10 km covered). None mentioned harvesting parts from wild populations. A high proportion of informants (63%) sold fruits and stem bark whereas a relatively low proportion of them (37%) sold fruits, stem bark, and leaves. The stem bark was recorded as most in demand followed by fruits. Respondents mostly confirmed the species scarcity. This species was used to treat 13 diseases and disorders. The stem bark was the most cited in the management of stomach infections and gynecological disorders. Fruits were mainly used in magic rituals and the treatment of stomach infections. Five preparations were recorded whither 54% of traders mentioned bark decoctions and 27% highlighted infusion of fruits in water. Overall, *Kigelia africana* is an important plant in Beninese ethnomedicine and the harvest and trade of its different parts represent major threats. Therefore, urgent conservation tools and actions are needed.

**Keywords:** *Kigelia africana*; ethnobotanical knowledge; Benin; herbal medicine traders; infections



## 1. Introduction

Due to the importance of medicinal plants to human life, it is necessary to investigate the ethnobotanical knowledge related to the medicinal and medico-magic utilization of plant species. Cunningham [1] reported that 70 to 80% of the world population uses plants for their primary healthcare. Furthermore, plant species harvested from wild populations serve as raw materials for commercial pharmaceutical factories and for local informal trade [2]. Medicinal plants are globally valuable sources of herbal products, meaning that people are worldwide using them both in folk medicine and pharmacological studies leading to various discoveries on their properties but many wild populations where plants are harvested are under threat and some are disappearing at a high speed [3]. The same authors stated that more than one-tenth of plant species are used in drugs and/or health products. Consequently, some authors argue that sustainable harvest is the most important conservation strategy for wild-harvested medicinal species, contributing to local economies and their long-term value to harvesters [4]. Medicinal plants are receiving research attention in many parts of the world. For instance, it has been reported by Chen et al. [3] that China and India have the highest numbers of medicinal plants used, with 11,146 and 7500 species, respectively. Furthermore, regarding the interest in the historical and current

use of medicinal plants by African populations, there is still a need to undertake research not only on the medico-magic exploitation of plant species but also on the impact of plant exploitation on their sustainable conservation. According to van Andel and Fundiko [5], maintaining cultural identity and resorting to herbal medicine in case of illness motivates migrants to continuously exploit medicinal plants of their origin during their stays in Europe and the United States. In Africa, as elsewhere, medicinal plants are collected from various types of plant communities. For example, plants are harvested from sacred groves in India [6] and from termite mounds in Benin [7] for medicinal and medico-magic utilization. Furthermore, the medicinal exploitation of mangrove phytodiversity was recently reported by Dossou-Yovo et al. [2]. Many studies are carried out on medicinal species throughout the world but there is still a need to focus studies on important species, of which some are threatened [8,9]. This matches with the fact that in West Africa, researchers investigated many medicinal species but the gap persists on some of them [10,11].

*Kigelia africana* (Lam.) Benth, commonly called sausage tree, belongs to the Bignoniaceae botanical family. It is a tree with low branches, up to 10–12 (20) m high which is established in Sudanian and Guinean savannahs and in some semi-deciduous forests, mostly on well-drained lands [12]. It is a medicinal plant species on which much attention should be focused. In fact, in the African pharmacopoeia, *K. africana* is used in treating different pathologies such as wound healing and microbial infections [13]. Moreover, diabetes treatment was stated through the pharmacological characteristics of extracts of parts of *K. africana* as well as the trace element and the chemical composition of this species extracts have recently been documented by Fagbohun et al. [14–17]. Singh et al. [18] also reported the organs of *K. africana* in treating skin diseases and other pathologies. In Zimbabwe, bark of the species is used to relieve toothache [19]. The antioxidant, anti-inflammatory, and anti-cancer activities of extracts of the plant were recently reported [20,21]. Therefore, there is evidence of the importance of *K. africana* in pharmacology and traditional medicine, and despite much scientific effort, there is a need to conduct more research and to scientifically validate other traditional uses of *K. africana* [22]. The species has recently been recorded among plant species used to treat infertility of women in Benin [23] where it is also widely used to treat diabetes [24]. Despite the widespread sale of *K. africana* in the market and its known uses, this species has received very little attention from researchers in Benin. However, market integration is one of the main threats to medicinal species [25]. Moreover, a recent study recorded *K. africana* as a threatened species in Benin [24]. Adverse weather effects on protected areas combined with other forms of stress, including anthropogenic effects such as over-consumption (bark of *Kigelia africana*), and the pollution of urbanization threaten the conservation of *K. africana* in Benin and other west African countries [26]. Considering this conservation status of the species in Benin, its importance in local pharmacopeia and the importance of the sustainable use of the species, there is a need to know more about its use profile. Thus, the aim of this research was to assess the use profile of *K. africana* through an ethnobotanical market survey in Benin. This study will contribute to acknowledging the importance of *K. africana* among local populations as well as the scarcity or availability of the plant organs in recent years.

## 2. Material and Methods

### 2.1. Study Area

Surveys were undertaken from January to March 2020 in some of the most populated towns of southern Benin, and with herbal medicine traders in these markets. These were the Pahou, Zobê, and Kpassê markets in Ouidah District, with 445 inhabitants/sq km (the Atlantic Department), the Cococodji market in the Abomey-Calavi District, 1010 inhabitants/sq km (the Atlantic Department), and Vêdoko and Dantokpa markets in Cotonou district with 8595 inhabitants/sq km (Department of Littoral, Cotonou, Benin).

*2.2. Data Collection*

Our study proposal was submitted to and approved by the head of our research institution for ethics for ethnobotanical investigations through market surveys. Indeed, the study protocol as well as the full manuscript received the ethical clearance of the Scientific Council of the University of Abomey-Calavi (N° 145-2021/UAC/VR-RU/SCS/SA). The aim of our research was explained to the responsible of each market in order to obtain the approval to conduct surveys and all market responsibles that we met gave their verbal approval. Similarly, the aim of the study was explained to each trader to get the verbal consent to participate in the research and this consent was obtained from all participants. There is a need to explain that the verbal, not signed, approval was considered in the frame of this research since it does not involve any use of human organs or tissues. In six markets, based on a semi-structured questionnaire, the ethnobotanical uses of *K. africana* were investigated. In each of the six surveyed markets, six traders were randomly selected with no distinction of ethnic group and age. As mentioned above, all of them gave their consent, meaning 100% acceptance to participate in the research. They were then interviewed based on a printed semi-structured questionnaire (available from authors). Thus, 36 herbal traders were consulted on the marketing of *K. africana* plant organs as well as advising on the use of parts of this species to multiple customers. The main questions concerned the age and ethnic group of the herbal trader, the trade of the parts or not, the diversity of the species parts they use to sell, the sources of the species parts they sell, the ranking of the species organs according to the demand, and the diversity of uses known for each organ. In addition, the scarcity versus availability of the species parts in the recent years were documented as well as the causes of scarcity when reported. The aim behind the question dealing with the trade of the species parts or not was not to exclude those who would report not trading them but to find out the importance of the species after surveying the total number of respondents in the frame of this study. Informants who would report not trading the species parts will be asked to mention the likely reasons. To better recognize the species organs during the survey, we bought and kept fresh and dry stem barks and leaves although they are very common in southern Benin and well known to the authors. It is important to highlight that the fruit of *K. africana* is well known to the authors and in southern Benin and easy to recognize no matter how fresh or dry it is.

There is evidence of the diversity of categories of people involved in the utilization of the parts of *K. africana*. However, since our study deals with investigating the use profile of the research species through a market survey, we collected data from herbal medicine traders and species not herbal medicine consumers nor traditional healers. In fact, herbal traders are known as involved in the trade of the species parts and many other medicinal plants since time immemorial. All methods were performed in accordance with the relevant guidelines and regulations for ethnobotanical and ethnobiological studies.

*2.3. Calculations and Statistical Analyses*

Various percentages of traders who mentioned species parts and diseases were calculated using the formula

$$Percentage\ of\ traders = \frac{Number\ of\ respondents\ mentioning\ X}{Total\ number\ of\ respondents} \times 100$$

We tested the influence of socio-demographic characteristics of the respondents on the ethnomedicinal knowledge they hold regarding *K. africana* by performing a simple linear regression at 0.05 level of probability using Stata15. The correlation between variables was also assessed at 0.05 level of probability. The demographic parameters were sex, age, ethnicity, profession, years of experience as herbal trader, and level of education (illiterate, primary level, secondary level, and university level) while the ethnomedicinal knowledge concerned the number of *K. africana* uses each trader reported.

## 3. Results

### 3.1. Gender, Age, and Profession of the Respondents

All the surveyed informants were women living in southern Benin, especially in the surveyed districts. The 36 informants, randomly selected, confirmed the trade of the species parts. The mean age of these traders was 46 years old with a range of 17 to 75, and, in general, they belong to five ethnic groups, which are Fon (53% of respondents), Aïzo (8% of respondents), Xwla (3% of respondents), Pedah (11% of respondents), and Yoruba (25% of respondents). However, all informants mastered Fon, the most spoken local language in southern Benin. All of them confirmed that they are fully involved in herbal medicine trade as a profession. The majority of informants had primary school level education (42%) while 31% reported a secondary school level and 27% of them were illiterate. Half of them reported 10 to 20 years of experiences as herbal traders, 19% have been selling medicinal plants since 20 to 30 years, whereas 17% hold less than 10 years and 14% of them have been trading medicinal plants for over 30 years. The simple linear regression between the socio-demographic variables and use reports on *K. africana* revealed that these variables did not influence the number of medico-magic uses that a trader holds on this species ($p > 0.05$; Table 1). Similarly, there was no correlation between the different variables ($p > 0.05$; Table 2).

**Table 1.** Results of the simple linear regression.

| Variables | Coefficient | Probability |
|---|---|---|
| Age | 0.0207914 | 0.490 |
| Ethnicity | −0.2716058 | 0.367 |
| Years of Experience | 0.0352425 | 0.369 |
| Education Level | 0.5043646 | 0.444 |
| Number of Uses | 3.454395 | 0.072 |

**Table 2.** Correlation between variables.

| Sex | Sex | Age | Ethnicity | Years of Experience | Education Level | Number of Uses |
|---|---|---|---|---|---|---|
| Sex | - | | | | | |
| Age | - | 1.0000 | | | | |
| Ethnicity | - | −0.1560 | 1.0000 | | | |
| Years of Experience | - | 0.4381 | −0.1143 | 1.0000 | | |
| Education Level | - | −0.6248 | 0.1659 | −0.2370 | 1.0000 | |
| Number of Uses | - | 0.1601 | −0.1767 | 0.2243 | −0.0004 | 1.0000 |

### 3.2. Sources of Kigelia africana Parts Sold by Medicinal Plant Traders

Table 3 shows that 36% of respondents confirmed that they purchase *K. africana* plant parts from the same markets where they sell them and/or by travelling far, between 3 to 10 km, to make the purchases. The same proportion of traders mentioned that they travel at least 25 km to obtain their goods from rural areas as well as from other large markets. A relative minority indicated that they obtained these plant organs only from the market where they sold them, and a very small proportion said they obtained plant organs from local and nearby markets.

It is important to note that none of the plant traders reported harvesting the species from the wild despite the proximity of some of the markets surveyed to old fallow lands and forests accessible to local people.

**Table 3.** *K. africana* parts, source, sold, local names, and traders' perception from the interviews.

|  | Number of Respondents (N = 36) | % or Rank |
|---|---|---|
| Source of *K. africana*: | | |
| - In their market and travelling far | 13 | 36 |
| - Only travelling very far | 13 | 36 |
| - Exclusively in their market | 6 | 18 |
| - In their market and nearby market | 4 | 10 |
| Main Sold Parts | | |
| - Bark (Yamblikpogoto) | 36 | 1st |
| - Fruits | 36 | 2nd |
| - Leaves | 36 | 3rd |
| Parts Sold | | |
| - Only fruits and bark | 23 | 63 |
| - Fruits barks and leaves | 13 | 37 |
| Mostly Sold Parts: | | |
| - Bark | 23 | 63 |
| - Fruits | 8 | 23 |
| - Bark equal to fruits | 5 | 14 |
| Parts' Local Names | | |
| - Bark: Yamblikpogoto (Fon) | 19 | 53 |
|    Ekpakpahoundoror (Yoruba) | 9 | 25 |
| - Fruit: Anonkan or Yamblikpo sin atississin (Fon) | 19 | 53 |
|    Kpahoundoror (Yoruba) | 9 | 25 |
| - Leaves | 36 | 100 |
| - Whole tree: Yamblikpo (Fon) | 19 | 53 |
|    Iguikpahoundoror (Yoruba) | 9 | 25 |
| Traders' Perception: | | |
| - Parts very scarce | 23 | 64 |
| - Parts available throughout the year | 6 | 18 |
| - Parts available in rainy season | 6 | 18 |

### 3.3. Ranking and Availability of Kigelia africana Parts Sold by Traders

Table 3 reveals that stem bark, fruits, and leaves were recorded not only as the main sold but also the most commercialized parts of *K. africana* plant.

As shown in Table 3, the vast majority of the informants sells only the fruits and stem bark, while a minority sells fruits, stem bark, and leaves of *K. africana* plant. However, none of the traders sell only one part of the plant. Table 3 reveals a majority of the interviewees citing the stem bark as the organ most in demand, followed by fruits. Stem bark and fruits are cited by a minority as being sold in equal parts, while leaves are cited as the least sold part. Figure 1 shows photos of all these organs.

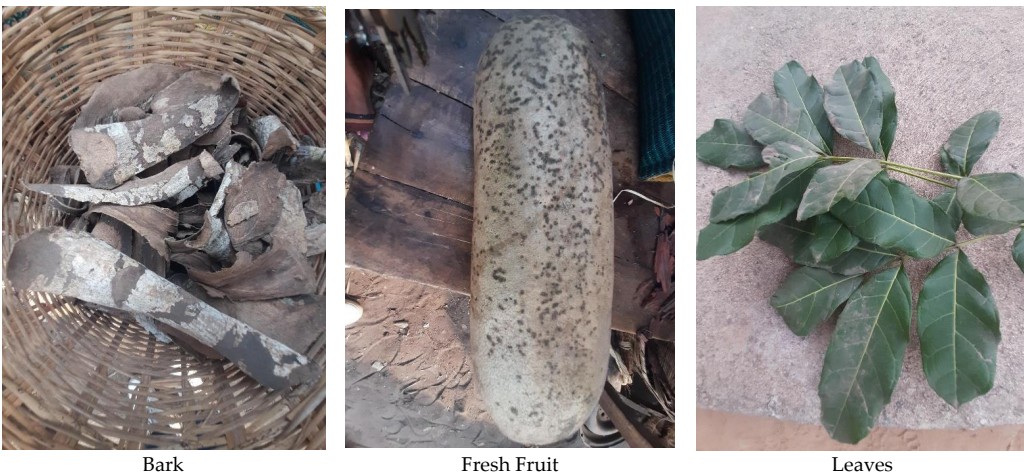

Bark                    Fresh Fruit                    Leaves

**Figure 1.** Different traded parts of *K. africana* plant.

Based on Table 3, most of traders highlighted that *K. africana* parts available for sale have become very scarce in recent years and a low proportion of them reported the availability of the species throughout the year as well as availability during the rainy season.

### 3.4. Diversity of Diseases, Disorders, and Rituals Treated Using the Species Parts According to Traders

Globally, 13 diseases, disorders, and magic rituals were recorded. While eight diseases and disorders are treated with the stem bark, seven diseases, disorders, and magical rituals are treated with fruit. Leaves are reported to be used only in magic rituals. Generally, the use of stem bark or fruit is specific in curing gynecological disorders and leg infections. In addition to these diseases, the bark of *K. africana* serves to treat stomach infection, stomachache, anemia during women's menstrual periods, vomiting, cough, and constipation. The greatest percentage of herbal traders (72%) mentioned the use of stem bark to treat infections followed by gynecological disorders (18% of respondents). Regarding fruit utilization, apart from the two common diseases, treatment of hemorrhoids and obesity, stimulation of breast milk production, wound healing, and magic rituals were all reported. The use in magic rituals is locally called "*Vossissa*" or "*Vodounsakpata do ayi*" in Fon and deals with laying out African traditional god called *"Sakpata"*. People put fruits of *K. africana* on firewood gathered in the field or bush to deter others from stealing it. Regarding the importance of fruit use, 27% of herbal traders mentioned the use of fruits in magic rituals followed by stomach infections reported by 18% of them.

### 3.5. Diversity of Preparation Modes According to Herbal Traders

In order to treat the diseases and disorders and for ritual uses, five types of preparation were reported to treat various pathologies, building decoction, infusion in a traditional local alcoholic beverage locally called "*Sodabi*" in Fon, infusion in water, the direct use of the entire part, and its use as a powder. The first two modes are used for the stem bark whereas the remaining modes as well as decoction are used for fruits. Thus, decoction was the only common preparation mode recorded for bark and fruits. The leaves were evoked as an integral part of the rituals. With regard to the importance of each type of preparation, the greatest percentage of herbal traders (54%) mentioned the use of bark as decoction while infusion of fruits in water was reported by 27% of respondents followed by decoctions and use of the entire plant in rituals (18% of informants for each). It should be noted that the fruit is ground and used to bandage lesions, which was reported by 9% of the medicinal plant traders surveyed.

## 4. Discussion
### 4.1. Gender, Age, and Profession of the Respondents

The mean age of the surveyed traders confirmed that herbal medicine trade is an activity mostly undertaken by people of a considerable age and holding a certain life experience in the field of traditional healing. The number of ethnic groups recorded for traders revealed that the herbal medicine trade has been undertaken by a wide range of communities for decades. In fact, many African indigenous communities depend on this commerce for their survival since time immemorial. Traders from various ethnic groups hold ethnomedicinal knowledge on *K. africana*, meaning that the species plays a great role and more conservation strategies are required for its populations. Based on the 36 questioned traders, we noticed that the sex, ethnicity, age, and total years of experience of herbal traders did not influence the number of uses that each trader holds on the study species. Although traders were randomly selected in this study, the non-influence of age on the total uses confirms the fact that both children and adults learn about the use of the medicinal plants mainly from their family [27]. In addition, since young and old traders are together in markets, there is a permanent knowledge exchange between them. This allows to confirm that there may be other socio-economic and demographic characteristics of respondents determining the number of uses that they know of for *K. africana*. Apart from

this preliminary market survey on the use profile of *K. africana*, the authors suggest further investigations on the diversity of ethnobotanical knowledge held by a greater number of respondents belonging to a diversity of ethnicity on the species throughout Benin. Since all respondents confirmed the herbal medicine trade as their profession, there is a need to better define conservation strategies towards medicinal plants and especially *K. africana* to sustain this longstanding activity in Benin, Africa, and other parts of the world for the well-being of herbal traders and customers. Herbal medicine traders were considered in this study and further studies on the use profile of *K. africana* in Benin can target other categories of informants such as herbal medicine consumers, as well as traditional healers in order to gather a diversity of knowledge related to this species exploitation. Moreover, outputs of such investigations will serve to highlight the similarity or difference in the medicinal utilization of the research species according to the target groups.

*4.2. Sources of Kigelia africana Parts Sold by Herbal Traders*

The trade of the species parts by all the randomly selected respondents proves the importance of *K. africana* in folk medicine in Benin. In other words, parts of the species are demanded by herbal medicine consumers to treat a diversity of diseases.

A large part of the traders are obliged to travel long and very long distances to acquire the plant organs. Although *K. africana* is both a wild and cultivated species in Benin, this long travelling of traders confirms the critical scarcity of the species in the last few decades reported by them. It also corroborates the fact that none of the surveyed traders mentioned collecting the plant from wild populations. According to Chen et al. [3], medicinal plant resources are being harvested in increasing volumes, largely from wild populations. Therefore, if *K. africana* were readily available in the wild, traders might be expected to collect its parts free of charge for sale. Moreover, some surveyed markets are close to natural forests and old fallows. In addition, the low proportion of herbal traders who mentioned that they are exclusively provided with the species parts in their own markets is a real proof of scarcity. However, the limitation of this research regarding the sources is that some sellers in the markets may actually harvest this plant from the wild populations but did not admit it since it is scarce. It is important to highlight that *K. africana* is known as both a wild and cultivated species in Benin [26], and the fact that no trader reported purchasing the species parts from other people's houses allows us to assume that the species is not only scare in the wild but also less and less cultivated. This assumption corroborates findings of recent study reporting *K. africana* as a threatened species in Benin [24]. The authors suggest more investigations especially towards harvesters and wholesalers of medicinal plants to better know the sources of the species organs they sell. Such studies will give a greater insight into the species scarcity and help define better sustainable strategies and tools for the species conservation.

*4.3. Ranking of Kigelia africana Parts Sold by Traders*

Many traders sold fruit and bark of the species and in a relatively small proportion. The organs of *Kigelia africana* were often claimed to be scarce and trade was concentrated on the bark, followed by the fruit. The bark harvesting may be a serious damage to the conservation of the species. In fact, Guidigan et al. [26] mentioned the over-consumption of bark as a threat to *K. africana* in Benin. Additionally, the fruits and leaves collection, as well as the flower cutting may sensitively reduce the reproductive capacity of the species. These findings confirm that urgent conservation of the study species is required. In fact, Vodouhè et al. [28] confirmed that non-timber forest products cannot be sustainably harvested in the absence of yield studies, harvesting adjustments, and monitoring of regeneration. Therefore, there is a need to conduct such investigations on *K. africana* in Benin to contribute to its sustainable conservation. Indeed, years ago, Gaoue and Ticktin [29] noticed a negative impact of bark and foliage harvest on the reproductive performance of *Khaya senegalensis* in the country. Documenting the impact of bark and foliage harvest on the reproductive capacity of *K. africana* in Benin is required to play a key

role in the species conservation. Compared to practices in South Africa, bark is ranked third among the most frequently sold parts of a range of medicinal plants, after roots and bulbs [24]. Similarly, Dossou-Yovo et al. [7], examining the utilization of medicinal plants harvested from termite mounds in Northern Benin, found bark as the most commonly used, followed by leaves. Similar to the present findings related to bark as the most demanded, Gaoue and Ticktin [29] stated that the bark of *K. senegalensis* plays a decisive role in providing medicines to the indigenous populations of Benin. Mahunnah [30] reported that medicinal plants are important worldwide and the conservation problem is a major threat to the rich biodiversity of medicinal plants in Africa. Moreover, many years ago, Nigro et al. [31] highlighted that the material from which useful compounds are extracted is not always cultivated or wild-harvested in a sustainable way. These allow us to assert that the collection of bark, fruits, and leaves of *K. africana* plant endangers the species conservation since there is no control under such collections. In addition, the majority of herbal traders reported, while dealing with the availability versus scarcity of the species parts, that logging for construction and climate change are the major threats to the conservation of *K. africana* in its natural ecosystems. These precisions, in addition to the exploitation of the stem bark, reveal threats to the species so the conservation of *K. africana* should be seen as a priority.

### 4.4. Diversity of Illnesses Disorders Treated and Rituals According to Traders

Thirteen diseases, disorders (being temporary dysfunctions of the human organism), and magic rituals were recorded in this study. This corroborates the findings that African people have appropriate knowledge on medico-magic uses of plant species. In other words, the connection between African populations and the uses of medicinal plants to cure illnesses dates from the far past. Dossou-Yovo et al. [7] reported 30 affections and disorders treated using 22 plant species; the present study revealed 13 illnesses and disorders cured, and rituals performed using parts of just one plant species, *K. africana*. This finding suggests the great importance of *K. africana* in the local populations' life. *K. africana* is used in African folk medicine and also in Indian traditional medicine, showing convergence in the uses of plant parts and diseases treated with it [18,19,32]. This study focused only on knowledge of *K. africana* held by the herbal medicine traders; there is a need to conduct other investigations concerning the species use profile from herbal medicine consumers and traditional healers belonging to various ethnic groups throughout the country. Such studies will lead to better stating the use profile of the research species.

### 4.5. Diversity of Preparation Modes

Five types of preparation were recorded in this study, proving the rich diversity of medicinal and medico-magic knowledge of *K. africana* plant according to traders. According to Yang and Ross [33], out of all the traditional recipes in China, decoctions show the highest efficacy of action and their preparation requires a reasonable amount of time. When investigating the ethnopharmacological uses of seven medicinal plants in Mali, Togola et al. [34] reported the decoction of leaves as the main form of preparation, and leaf powder was mostly used for infusions. Similarly, the present study stated decoctions as the predominant type of preparation, indicating the efficacy of decoctions in herbal medicine. The infusion of fruit, which is the second most important in our study, reinforces previous results which stated decoction and infusion as the main and most effective forms. Indeed, Jackson and Beckett [19] reported, from Zimbabwe, the use of decoctions of bark of *K. africana* to treat dental pain, as well as infusions of its bark and crushed fruit in the treatment of many diseases. Therefore, the decoction of stem bark of *K. africana* plant plays a crucial role in phytomedicine but, as mentioned above, the harvesting of this organ threatens species conservation.

## 5. Conclusions

*Kigelia africana* is highly requested for phytotherapy in southern Benin. Parts of the species plant are sought after either for their quality and therapeutic effectiveness or their abilities to confront many diseases, disorders, and mystical problems. The harvesting and trade of the plant parts threaten the survival of the tree. Thus, there is a need to undertake many other studies on the species in order to define specific conservation actions towards *K. africana* through the germination and planting of the species throughout Benin. In other words, there is a need to strengthen forestry services capacities, consolidating technical achievements and monitoring forestry actions for the conservation of *K. africana* in Benin. This will contribute to the long-term sustainable exploitation of the species parts as medicine.

**Author Contributions:** H.O.D.-Y. was the originator of the research. He wrote the first draft of the research proposal, participated in the data collection and analysis, as well as manuscript writing and its submission. F.G.V. participated in proposal writing, collaborated during data collection, and contributed to analysis, manuscript writing, and editing. V.K. participated in the manuscript reading and editing. B.S., as head of the laboratory, read, edited, and commented on the manuscript for its improvement. He provided valuable comments that helped improve the quality of this manuscript. He read and gave approval of the revised version of the manuscript before its submission. All authors reviewed the manuscript. All authors have read and agreed to the published version of the manuscript.

**Funding:** This research received no external funding.

**Institutional Review Board Statement:** The study was conducted in accordance with the Declaration of Helsinki and approved by research institutes. In fact, prior to data collection, our study proposal was submitted to and approved by the Head of our research institution, Laboratory of Applied Ecology of the Faculty of Agronomic Sciences of University of Abomey-Calavi, Benin who participated in this research, for ethics for ethnobotanical investigations through market surveys. The study proposal in a project protocol received the ethical clearance of the Scientific Council of the University of Abomey-Calavi (N° 145-2021/UAC/VR-RU/SCS/SA on 4 April 2021).

**Informed Consent Statement:** The aim of our research was explained to the responsible of each market in order to obtain the approval to conduct surveys and all market responsibles that we met gave their verbal approval. Similarly, the aim of the study was explained to each trader to get the verbal consent to participate in the research and and publish this paper. Informed consent was obtained from all subjects involved in the study.

**Data Availability Statement:** The ethical clearance as well as data supporting various calculations and analyses are made available at Conservation MDPI (https://www.mdpi.com (accessed on 14 April 2022)).

**Acknowledgments:** Authors are grateful to all market responsibles for giving the approval to undertake this research. They thank herbal medicine traders who participated in this research. They deeply thank Phil Harris from Coventry University for reviewing the first draft of this work, and anonymous scientists who reviewed this work. They helped improve its scientific quality; our gratitude to Research Square for posting the draft of this manuscript.

**Conflicts of Interest:** The authors declare no conflict of interest.

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
