# Peer review of "Investigating the Use Profile of Kigelia africana (Lam.) Benth. through Market Survey in Benin"

_conservation, doi:10.3390/conservation2020019_

Round 1
Reviewer 1 Report
Reviewer report
Manuscript title: Investigating the Use Profile of Kigelia africana (Lam.) Benth. through Market Survey in Benin.
I found this work very interesting, because it draws attention to a threatened species (Kigelia africana) known for its therapeutic virtues in Benin.
Questions
- Why is this species chosen by the authors?
- Why only traders were questioned in this study?
- Kigelia africana is wild or cultivated species? Is this species support salinity, drought, etc.? This species is threatened by irrational harvesting or climate change?
- The bark cited is fruit bark, stem bark or root bark? The knowledge, of which part bark is harvested and used, may help us understanding why this species is threatened and why it is necessary to conserve it.
- The difference between disease and disorder?
Abstract
Line 10: “but receives little attention” from whom?
Line 10: to replace “Indeed, the species” by “In addition, this species” Because “indeed” usually used to confirm information cited before “indeed”
Line 11: more details are required concerning the methodology of this work like semi structured questionnaire is done by distribution of printed questionnaire or face to face or what, The number of questioned traders, the period during which this survey was carried out.
Line 12: it is possible to indicate the number of questioned population in the result part like: Among 36 questioned herbal traders, 36% of respondents obtained K. africana parts by…
Line 14 and 15: I can understand the difference between the most sold “fruits and bark” and less sold “fruits, bark and leaves”? In addition, it is better to mention the percentage of respondents between parakeets (just my opinion).
Line 17: This species was used to treat 13 diseases and disorders. The bark was the most cited in the management of …
Line 19: to replace “and 54%” by “whither 54%”
Line 20: “, 27%” to replace comma by “and”
Key word: Kigelia africana, ethnobotanical knowledge, Benin, herbal medicine traders, infections.
The most important key word is the name of species so it is necessary to put it in first place.
Introduction
Line 26-27: to replace “Medicinal plants are so…” by “Due to the importance of medicinal plants for human life, it is necessary…”
Line 28: to add that after “reported”
Line 66: to replace “of extracts of the plant” by “of this species extracts” for avoiding repetition of “of” several times
Line 70: “The species” by “This species”
Line 72: “used” is repeated twice. The first one must be deleted.
Line 72-74: “K. africana has received very little attention in Benin where, due to its known uses, the species parts are widely sold in markets.” This sentence needs to be rectified “little attention” and widely? In my opinion, I think it is better to replace the sentence by “Despite the wide sold in market of K. Africana and its known uses, this species…” is stronger.
Material and methods
More details are required concerning the area of study, the period of survey, the age range of questioned traders, the selection method of traders (randomly or not) and the most important questions asked by authors in the questionnaire.
Results and discussion
- In the part of gender, age and profession of respondents: it is good to give more demographic information about participants in table citing: sex, age, family situation, educational level, ethnic groups and profession with percentage of each category. Additionally, the association between knowledge, use and source of the studied species and different demographic characteristics by calculating chi2 may help knowing the importance of this species in the studied area.
- Table 1: to delete “showing the” directly begin with “Proportion of…”
- The same remark like Table 1 for all tables.
- I find that it is better to merge all the tables in a single table contains all the information like this:
|
Number of respondents |
% |
Source of K. Africana: - In their market - … |
|
|
Main sold part - Bark (Yamblikpogoto) - … |
|
|
Part sold - … - … |
|
|
The mot sold part: - … - … |
|
|
Traders’ perception: - … - … |
|
|
- Figure 1: To delete “showing the” directly put “Traded organs of…”
- Concerning the diseases treated by K. Africana: I advise the authors to put a table containing the following data as showing the table below
Disease, disorders and ritual |
Part used |
Preparation mode |
Administration route |
Treatment duration |
Active ingredient (with reference) |
Number of respondents |
% |
|
diseases |
Stomach infection |
|
|
|
|
|
|
|
|
|
|
|
|
|
|
|
|
|
|
|
|
|
|
|
|
|
Disorders |
|
|
|
|
|
|
|
|
|
|
|
|
|
|
|
|
|
ritual |
|
|
|
|
|
|
|
|
|
|
|
|
|
|
|
|
The presentation of K. Africana uses like mentioned above gives more importance and value to this work and the studied species.

Author Response
Our sincere thanks for your valuable comments and suggestions.
Best wishes
Hubert. O Dossou-Yovo (MSc)

Reviewer 2 Report
The manuscript “Investigating the Use Profile of Kigelia africana (Lam.) Benth. through Market Survey in Benin.” shows data on the ethnobotanical use of the species Kigelia africana (Lam.) Benth. in Benin, where it has been recorded as threatened. The authors seem to be motivated to highlight the importance of the plant to local populations as well as its scarcity in order to encourage its conservation, which is a noble goal. The data were collected through semi-structured questionnaires from 36 herbal medicine traders from six markets (six traders per market). The manuscript is well organized and well written, but some issues need to be addressed.
Major remarks:
The aim of this research was to assess the use profile of Kigelia africana. Why did you decide to interview (only) traders of K. africana plant organs and not the consumers? How can you be sure that plant parts are used and prepared in a certain way if the answer was not obtained from the final consumers? Would it be more accurate to say that the traders recommended the said use to the consumers?
Could you provide more detail on the participants of the study? How did you choose the six traders that were interviewed per market? Were all traders marketing K. africana or did you exclude those not marketing it from the study? If so, how many did you ask to participate in the study and what was the % of acceptance? Why only verbal approvals/verbal consents were collected?
Please explain what “travelling far” and “travelling very far” implies (rows 117−119).
In the Discussion, the authors state that many traders are obliged to travel long and very long distances to acquire plant organs (rows 195−196), i.e. to buy them from other traders. Is it known how the latter acquire the plant parts? Can they only be collected from the wild or is the species also cultivated in Benin? Why were the people collecting the plant not included in the study? Don’t you think that they could give a greater insight into the species scarcity?
None of the respondents mentioned collecting the plant parts from wild populations, which, according to the authors, may not be true for all surveyed traders. The authors also hypothesize the bark use to threat the species conservation “since there is no control under such collections”. However, as can be seen from the manuscript this is just an assumption since they themselves did not establish the latter claim.
row 60 – could you include some examples for which pathologies is the species used? Later, you name them but as an addition to the first sentence (“Moreover,…”). Therefore, it would make sense to mention some at the beginning as well.
References in the text and in the reference list should adhere to Instructions to Authors. Also, there are not as many references as indicated in the reference list (38) since they are not formatted correctly (e.g. ref. 12 and 13, 22 and 23, 24 and 25, 27−29).
Other remarks:
row 3 – is the period at the end of the title necessary?
In the Abstract, the number of surveyed traders and the definition what is considered travelling “far” and “very far” could be added in parentheses.
Not all keywords need to be in capital letters.
Species name in tables, table headings and in the text (e.g. row 271) should be given in italics.
Minor writing mistakes in the text should be corrected (e.g. rows 56, 72, 132, 208, 228, 277, 278).
Author Response
Sincere thanks for your valuable review.
Best wishes
Hubert O. Dossou-Yovo (MSc)

Reviewer 3 Report
The present paper reports an investigation of the Use Profile of Kigelia africana (Lam.) Benth. through Market Survey in Benin. The subject of this study is very interesting. The manuscript is relatively well written and deserves to be considered for publication after some revisions.
Abstract
Line 20 : replace « Kigelia africana is important in local ethnomedicine; harvesting and trade of its parts are threats. So, urgent conservation tools and actions are needed. » with «Overall, Kigelia africana is an important plant in the Beninese ethnomedicine and the harvest and trade of its different parts represent major threats. Therefore, urgent conservation tools and actions are needed.»
Introduction
The introduction part of the manuscript well explains the background, objective of the study and it contains sufficient references. However, the first part is too long and contains too many details, therefore kindly shortening these paragraphs and adding new data on medicinal plant in Benin.
Results
Line 116 : Replace « Table 1 shows that 36% of respondents confirmed that they purchase K. africana plant 116 parts both in the markets where they sell them and after travelling far to make the purchases » with « Table 1 shows that 36% of respondents confirmed that they purchase K. africana plant parts from the same markets where they sell them and/or by travelling far to make the purchases »
Line 118
The folowing sentence need to be rewrite « It should be noted that the same fringe of traders mentioned how they only travel very long distances to buy the plant organs they sell. They travel at least 25 km before purchasing their goods from rural areas as well as to other large markets ».
You can write
« The same fringe of traders mentioned that they travel at least 25 km to obtain their goods from rural areas as well as from other large markets.
Title of the Table 1. Proportion of traders according to source of K. africana parts
Line 124 : K. africana should be italic. Please correct it in whole document.
Table 2. showing the main sold and most commercialized parts of K. africana1 according to traders.
Replace the title by « Table 2. Main sold and most commercialized parts of K. africana according to traders ».
Line 136 : replace «No trader sells only fruit, only bark or only leaves » with « However, none of the traders sell only one part of the plant »
Table 3. Proportion of traders according to parts sold.
Table 4. Ranking of most sold parts of K. africana.
Line 143 : Replace Figure 1. « showing the traded organs of K. africana plant » with « Figure 1. Different traded parts of K. africana plant »
Table 5. Replace «showing traders’ perception of parts availability in recent years » by « Traders' perception of plant availability in recent years».
Discussion
Discussion part is adequate and well explained the results.
Conclusion
Line 267 : delete s « threatens »
Author Response
Sincere thanks for your valuable review.

Reviewer 4 Report
The authors studied the Use Profile of Kigelia africana (Lam.) Benth. through Market Survey in Benin. Kigelia africana is an important medicinal species of pan-tropical distribution. The study seems exciting.
Introduction:
Line 50-52: Reconsider this sentence; it seems confusing.
Line 60-63: rewrite this sentence
I suggest the authors provide information about the threat status of this species in the introduction part. Why is this species threatened in West Africa?
Material and Methods:
The authors didn’t provide the questionnaire in MS. It would be better if the questionnaire is provided in the supplementary information.
Results
Dost the use profile information provided about the plant corroborates the previously published articles. It would be interesting to discuss the already published data about this plant by comparing the current findings.
# Cross check to italicize the plant name consistently throughout the MS
Author Response
Sincere thanks for your valuable review.
Best wishes
Hubert O. Dossou-Yovo (MSc).

Round 2
Reviewer 1 Report
document quality has been improved thanks and good luck
Author Response
Sincere thanks for reviewing our manuscript.Reviewer 2 Report
The manuscript has been significantly improved according to given remarks.
I suggest writing zeros in Table 1 where lacking (also check the first coef) and writing numbers without zeros in Table 3 (e.g., "6" instead of "06").
Some corrections are still needed:
row 68 - "It is a tree with low branches"
row 81 - remove one "used"
row 87 - species name in italic
rows 147-148 - maybe the last sentence can be omitted considering it was already stated in the previous rows
rows 163, 164 - space may be added "p > 0.05"; cite both tables either with capital or small letters
row 229 - ethnicity?
row 264 - scarce?
row 300 - "being temporary dysfunctions..."
Table 1 - write all words in capital (Variables, Coef) and in full since there is enough space; Ethnicity
Table 2 - write Sex in capital
Author Response
Sincere thanks for reviewing our manuscript.

Reviewer 3 Report
The manuscript has been well developed.
Author Response
Sincere thanks for reviewing our manuscript.
We are grateful to the reviewer 3 who spent his/her valuable on our manuscript. Thanks are also due to the Editor in charge of the publication process.
The reviewer 3 approved the revised version of the manuscript.
No corrections provided.